# Ciliate Microtubule Diversities: Insights from the EFBTU3 Tubulin in the Antarctic Ciliate *Euplotes focardii*

**DOI:** 10.3390/microorganisms10122415

**Published:** 2022-12-06

**Authors:** Sandra Pucciarelli, Daniela Sparvoli, Patrizia Ballarini, Angela Piersanti, Matteo Mozzicafreddo, Lucia Arregui, Cristina Miceli

**Affiliations:** 1School of Biosciences and Veterinary Medicine, University of Camerino, via Gentile III da Varano, 62032 Camerino, Italy; 2Unit of Microbiology, Department of Genetics, Physiology and Microbiology, Complutense University of Madrid, 28040 Madrid, Spain

**Keywords:** tubulin function, microtubule diversities, gene-silencing, protozoa

## Abstract

Protozoans of the Phylum Ciliophora (ciliates) assemble many diverse microtubular structures in a single cell throughout the life cycle, a feature that made them useful models to study microtubule complexity and the role of tubulin isotypes. In the Antarctic ciliate *Euplotes focardii* we identified five β-tubulin isotypes by genome sequencing, named EFBTU1, EFBTU2, EFBTU3, EFBTU4 and EFBTU5. By using polyclonal antibodies directed against EFBTU2/EFBTU1 and EFBTU3, we show that the former isotypes appear to be involved in the formation of all microtubular structures and are particularly abundant in cilia, whereas the latter specifically localizes at the bases of cilia. By RNA interference (RNAi) technology, we silenced the EFBTU3 gene and provided evidence that this isotype has a relevant role in cilia regeneration upon deciliation and in cell division. These results support the long-standing concept that tubulin isotypes possess functional specificity in building diverse microtubular structures.

## 1. Introduction

Microtubules are polymers involved in fundamental processes of the eukaryotic cell. They are crucial for motility, cell architecture, organelle translocations, chromosome segregation and cell division [1]. Besides this functional versatility, the structure of microtubules is the same. Excluding a few exceptions, they are made of 13 protofilaments composed of α/β-tubulin heterodimers that interact laterally to form the tube-shaped polymer [2]. Microtubules are also composed of a variety of microtubule-associated proteins (MAPs) [3] and may differ in sensitivity to low temperatures. Generally, microtubules from homeothermic animals (mammals and birds) and other organisms that live in temperate environments usually disassemble at a temperature below 4 °C unless protected by specific MAPs [4]. In contrast, MAPs-free microtubules from psychrophilic organisms maintain typical polymerization/depolymerization properties even at the freezing temperatures of their habitat [5,6].

The α and β-tubulins share 40% amino-acid sequence identity. Each tubulin monomer binds a guanine nucleotide, which is nonexchangeable when it is bound in the α subunit, or N site, and exchangeable when bound in the β subunit, or E site [7]. Both undergo a variety of posttranslational modifications [8]. In most eukaryotic organisms, both α- and β-tubulin consist of isotypes encoded by different genes, which are grouped into six classes in vertebrates [9]. Tubulin isotypes vary from each other mainly by differences in their amino acid sequences usually clustered within the last 15 C-terminal residues, which constitute the isotype-defining domain. This region of the protein lies on the exterior surface of the microtubule and is the binding site for several MAPs [10,11]. 

Both α- and β-tubulin undergo numerous post-translational modifications such as glutamylation, glycosylation, tyrosination, acetylation, acylation, palmitoylation and phosphorylation [12], and most of them occur in the carboxyl-terminal regions of tubulins, likely functioning as molecular codes for the recruitment of specific MAPs [13]. The post-translational modifications affect the overall dynamic properties of microtubules during interphase as well as in mitosis [14,15], and often involve stable subpopulations of microtubules. Tubulin isotypes themselves modulate microtubule dynamics and organization [16]; they are responsible for changes in cell sensitivity to anti-cancer drugs [17,18,19,20] and for growth and maintenance of neurons and neurological disorders in humans [21,22,23].

In ciliated protozoa, all microtubule functions (cell architecture, movement, chromosome segregation and cell division) are carried out in a single cell [24]. It has been estimated that these organisms assemble 17 distinct types of microtubules throughout their life cycle [24,25]. Prior to high-throughput genome sequencing, genetic diversity of tubulins was considered reduced in ciliates, i.e., tubulin pools are composed by a single α- and β-tubulin isotype; thus, this microtubule complexity was ascribed to specific post-translational modifications of microtubules [24]. However, upon macronuclear genome sequencing of several ciliates, including *Euplotes* species, several novel α- and β-tubulin genes have been characterized. Therefore, a multigenic tubulin family is a common feature not only in mammalian cells [26] but also in ciliates, suggesting that diverse tubulin isotypes may be responsible for the formation of functionally different microtubules. In support of this hypothesis, Pucciarelli et al., have previously demonstrated that the ciliate *Tetrahymena thermophila* uses a family of distinct β-tubulin isotypes to assemble subsets of microtubules with diverse functions [27].

In this study, we sought for additional evidence in support of this scenario by investigating the psychrophilic Antarctic ciliate *Euplotes focardii* (CCAP1624/34). This ciliate shows strictly psychrophilic phenotypes, including optimal survival and multiplication rates at 4–5 °C [28], lack of a transcriptional response of the Hsp70 genes to thermal shock [29], and modifications in the α- and β-tubulin genes [30]. This protist expresses five β-tubulin genes, named EFBTU1, EFBTU2, EFBTU3, EFBTU4 (GenBank ID: AAB31932, ACN61490, ACN61491, ACN61492, respectively) and EFBTU5 (GenBank ID: MJUV00000000.2), which have been identified by different approaches, including genome-wide studies [31,32,33,34]. Similarly to other ciliated protozoa, such as *T. thermophila* [35] and *Paramecium tetraurelia* [36], *E. focardii* possesses two β-tubulin isotypes (EFBTU1 and EFBTU2) with 98% of amino acid identity, whereas EFBTU3 and EFBTU4 show a sequence identity with the other two tubulins ranging between 84% and 89% [5,30]. Finally, EFBTU5 is the most divergent isotype with a sequence identity lower than 60% and is comparable to the so-called β-like tubulins in *T. thermophila*. Among these isotypes, the EFBTU3 has attracted major interest for the following unique properties in the primary structure: a- most of the unique amino acid substitutions are located at the GTP/GDP binding site, at the Taxol binding site and at the M-loop, which may represent adaptive changes of the protein conferring assembly competence and stability in the cold [5]; b- it has been demonstrated that the rare Cys281 of EFBTU3, located in the M-loop, contributes to the folding of this isotype, which requires additional factors besides purified CTT and CofA to attain its native structure [37]; c- the expression of EFBTU3 isotype is stimulated by deciliation and increases during the early stages of cilia regeneration in contrast with the EFTBU2 expression, which appears mostly expressed when cilia are regenerated [5].

Moreover, EFBTU3 as the vertebrate class-III β-tubulin displays a serine residue at position 239 that substitutes a cysteine, which is conserved in most of the β-tubulin isotypes [38,39], including the other *E. focardii* isotypes. Here, we show that EFBTU3 specifically localizes at the bases of cilia in *E. focardii* cells and regulates ciliary regeneration.

## 2. Materials and Methods

### 2.1. Euplotes focardii Culture Conditions

The strain TN1 used in this study was isolated from sediment and seawater samples collected in Antarctica in 1988 (Terra Nova Bay coastal waters) [28]. *E. focardii* cell cultures were grown in a cold room at 4 °C using the green algae *Dunaliella tertiolecta* as a food source. Cells from logarithmic phase cultures were synchronized by three consecutive treatments of starvation and re-feeding for three days [40]. 

### 2.2. RNAi-Based Gene Silencing Experiment

The silencing approach used to suppress the expression of EFBTU2 and EFBTU3 genes is adapted from [41,42,43], where cells were fed with bacteria engineered to produce double-stranded RNAs (dsRNAs) targeting the gene of interest and thus are responsible for the degradation of the corresponding mRNA in the cell cytoplasm. *E. focardii* cells were grown in artificial sea water and fed with the algae *D. tertiolecta*. During the process of RNAi induction, cells were adapted to *E. coli* as a food source. Specific fragments were chosen for the knock-down of the tubulin genes and they were obtained by PCR amplification using cDNA as a template. cDNA was generated from 2 µg of total RNA extracted from *E. focardii*, using RevertAidM-MuLV Reverse Transcriptase (Fermentas, Milan, Italy) according to the manufacturer’s instructions, and the 201 bp and 406 bp fragments for the EFBTU2 and EFBTU3 were amplified with primers listed in Appendix A, respectively. The PCR products were first cloned into the pGEM-T Easy vector (Promega, Milan, Italy) and subsequently sub-cloned into the L4440 vector (kindly provided by Prof. Hans Joachim Lipps, Institute of Cell Biology, University of Witten/Herdecke, Witten, Germany) to generate EFBTU2-L4440 and EFBTU3-L4440 vectors. The final constructs were then introduced into the RNase III deficient *E. coli* strain HT115 [F-, mcrA, mcrB, IN (rrnD-rrnE)1, lambda-, rnc14::Tn10 (DE3 lysogen: lacUV5 promotor-T7 polymerase)] for dsRNAs synthesis. A 1:100 dilution of an overnight bacterial culture was grown to an OD_600_ of 0.4. Expression of double-stranded RNA was induced by adding 0.8 mM IPTG (isopropyl-beta-D-thiogalactopyraniside) to the bacterial culture grown for 4 h to an OD600 of 1. The dsRNAs-containing bacteria were washed at least twice with sterile water, killed at 42 °C and resuspended in artificial sea water to an OD600 equal to 4. 500 µL/day of this suspension were added to 10 mL of *E. focardii* cell cultures in presence of the algae *D. tertiolecta*. The gene silencing experiment was performed for ten days.

### 2.3. Estimation of EFBTU3 Gene Expression by Quantitative Real-Time PCR (qRT-PCR) in Euplotes focardii EFBTU3-Silenced and Non-Silenced Cells

Total RNA was extracted from two 100 mL cultures of *E. focardii* fed with IPTG-induced bacteria containing the EFBTU3-L4440 vector (EFBTU3-silencing) or the empty L4440 vector (control 1), respectively. Total RNA was also extracted from two additional 100 mL cultures of *Euplotes* (control 2 and 3) and fed with the same bacteria mentioned above but not induced with IPTG, to prevent the expression of double-stranded RNAs. RNA extraction was performed after 10 days of RNAi-based gene silencing, with TRI reagent (Sigma, Milan, Italy) according to the user’s manual. The quality of RNA was examined by electrophoresis on a 2% agarose gel containing 2% of formaldehyde. Potential contamination of the RNA by genomic DNA was tested by PCR amplification of 100 ng of total RNA using primers specific for *E. focardii* EFBTU2 as housekeeping gene. The predicted fragment was not detected after 30 cycles, each consisting of 50 s denaturation at 95 °C, 30 s primers annealing at 54 °C and 30 s elongation at 72 °C. First-strand cDNA was generated from 2 µg of total RNA using RevertAidM-MuLV Reverse Transcriptase (Fermentas, Milan, Italy) according to the manufacturer’s directions. The resulting cDNAs were amplified by qRT-PCR. The qRT-PCR was performed on *E. focardii* cDNA using the SYBR green DNA-binding method. The EFBTU3 gene and the ribosomal SSU housekeeping gene of *E. focardii* (acc. no. GenBank EF094961) were distinguished by using the primer pairs Beta3_qRT-PCR and SSUrRNA_qRT-PCR listed in Appendix A. The values of the standard curves are listed in Appendix A. An amount of 12.5 μL SYBR Premix Ex *Taq* (2×) buffer (TaKaRa Biotech, Beijing, China), 5 pg of each primer and water were added to 100 ng of *E. focardii* cDNA to reach the final volume of 25 μL. The PCR parameters were initial denaturation at 95 °C for 10 min to activate the polymerase, followed by 40 cycles of denaturation at 95 °C for 15 s and annealing and extension at 60 °C for 60 s each. Following amplification, melting curve analysis of the DNA was performed at temperatures between 50 and 95 °C, with the temperature increasing at a rate of 0.5 °C/10 s. All PCR reactions were performed in a Multicolor Real-Time-PCR Detection System iCycleriQ (Bio-Rad, Milan, Italy). During the primer annealing/extension step, the increase in the fluorescence from the amplified DNA was recorded by using the SYBR Green optical channel set at a wavelength of 495 nm. The initial threshold value was set at 30 fluorescent units.

### 2.4. Deciliation of E. focardii Cells

*E. focardii* cultures of EFBTU3 silenced and non-silenced cells were separately concentrated by centrifugation at 1900 rpm for 10 min and the supernatant was removed. Cell pellets (containing approximately 5 × 10^5^ cells) were resuspended in 300 µL of artificial sea water and equal volume of ethanol 14% was added. Cells were vigorously shacked for a few seconds and then observed by bright field microscopy to determine the effectiveness of deciliation, which typically was 90% or more. Finally, 150 mL of artificial sea water were added to each deciliated cell suspension culture to prevent cell death caused by ethanol. The whole volume of each culture was equally distributed in Petri dishes to better follow cilia regeneration of silenced and non-silenced cells in the 24 h after deciliation.

### 2.5. Polyclonal Antibodies to Detect EFBTU2 and EFBTU3 Isotypes

A rabbit polyclonal anti-EFBTU2/EFBTU1 and EFBTU3 primary antibody coupled with a peroxidase-conjugated goat anti-rabbit IgG secondary antibody (1:5000, Biorad 1:1000 dilution; GE Healthcare, Milan, Italy) were used to specifically detect these two tubulin isotypes. These specific antibodies were produced against the C-terminus of the EFBTU2 and EFBTU3 isotypes, which represents the most variable region among *E. focardii* β-tubulin genes. The EFBTU2 antibody is also able to bind the EFBTU1 isotype since the C-terminus is identical to that of the EFBTU2; in fact, these two isotypes encode almost the same protein. Peptides from tubulin sequences were synthesized by Merriefield solid phase procedures and purified to HPLC, as described elsewhere [44]. These peptides included CEEEGEFDDEEEMDV (anti-EFBTU2) and CHTYEEGEGEFDDEDSEL (anti-EFBTU3) sequences. The first was coupled to KLH (keyhole limpet hemocyanin) through its C-terminal cysteine employing *m*-maleimido-benzoyl-N-hydroxysuccinimide ester, and the last was conjugated to BSA (bovine serum albumin) with glutaraldehyde [45]. Rabbit immunization and screening of specific antibodies (ELISA, Western Blot and indirect immunofluorescence) was done as previously described [46]. To confirm results, both antibodies were synthesized for a second time by the biotechnological company GenScript. The commercial mouse monoclonal anti beta-tubulin antibodies were from Sigma-Aldrich (Milan, Italy) (clone TUB 2.1).

### 2.6. Immunofluorescence Microscopy

*E. focardii* cells in logarithmic phase were washed, placed on a polylysine-coated (0.5 mg/mL) coverslip and permeabilized with 0.2% Triton X-100 in PHEM buffer (60 mm Pipes, 25 mm Hepes, 10 mm EGTA, 2 mm MgCl2, final pH adjusted to 6.9 with NaOH). Cells were fixed with 2% paraformaldehyde in PHEM for approximately 30–60 min, washed once with NaCl/Pi (130 mm NaCl, 2 mm KCl, 8 mm Na_2_HPO_4_, 2 mm KH_2_PO_4_, pH 7.2) and then twice with NaCl/Pi plus 0.1% BSA; washes were 10 min each. Cells were incubated with primary antibody (Sigma mouse monoclonal anti-β-tubulin DM1B) alone and in combination with rabbit polyclonal anti-EFBTU2/EFBTU1 and rabbit polyclonal anti-EFBTU3 or with the GenScript antibodies overnight at 4 °C, washed and then incubated with secondary antibody (Alexa Flour 488 goat anti-(mouse IgG)/Alexa Flour 594 goat anti-(rabbit IgG), Invitrogen) for 1 h at 37 °C. Finally, cells were washed and suspended in 0.5% propyl gallate in glycerol.

Images of *E. focardii* cells were collected using an MRC600 Bio-Rad confocal system connected to a Nikon inverted fluorescence microscope (Diaphot-TMD equipped with an Apoplan60 objective; Nikon, Tokyo, Japan). 

### 2.7. Purification of Cellular Subfractions from E. focardii Cells

Cilia were obtained by cell deciliation, as previously described for the EFBTU3 silenced cells and non-silenced cells. After the treatment with the ethanol 14%, cells were centrifuged 5 min at 850 g to separate the supernatant containing cilia and the pellet containing cortical and cytoplasmic microtubules. Cilia were recovered by centrifuging the supernatant 30 min at 17,000× *g*. The pellet was re-suspended in PHEM buffer (60 mm Pipes, 25 mm Hepes, 10 mm EGTA, 2 mm MgCl_2_, final pH adjusted to 6.9 with NaOH) and sonicated to disrupt cell cortex and recover basal bodies. The sample was finally centrifuged 2 min at 14,000× *g* to separate the basal bodies in the pellet from the cytoplasmic microtubules in the supernatant.

### 2.8. SDS-Polyacrylamide Gel Electrophoresis (PAGE) and Immunoblotting

Proteins in each cell fraction (whose concentration was estimated by Bradford test) were separated by 8% polyacrylamide gel and transferred onto nitrocellulose sheet (0.45 µm, GE Healthcare, Milan, Italy) using standard conditions [34]. The membrane was incubated with the rabbit polyclonal primary antibodies (1:200 dilution; Invitrogen) and a peroxidase-conjugated goat anti-rabbit IgG secondary antibody (1:1000 dilution; GE Healthcare, Milan, Italy). Bound secondary antibody was detected by enzyme-coupled luminescence (Rodeo™ ECL Western Blotting Detection Kit, USB). Subsequently, the nitrocellulose membrane was stripped, as described in Pucciarelli et al. [34], for reprobing with a mouse monoclonal anti-β-tubulin primary antibody (1:200 dilution; Sigma, Milan, Italy) and a peroxidase-conjugated goat anti-mouse IgG secondary antibody (1:10,000 dilution; USB).

The same cell fractions were additionally separated on an 8% polyacrylamide gel and transferred onto nitrocellulose sheet, as described above, for incubation with a mouse monoclonal anti-γ-tubulin (1:200 dilution; Sigma, Milan, Italy) and a peroxidase-conjugated goat anti-mouse IgG (1:10,000 dilution; USB) secondary antibody. Bound secondary antibody was detected as described above.

A rabbit polyclonal anti-human-γ-tubulin primary antibody (1:1000 dilution) (kindly provided by R. Melki, CNRS, Gif-sur-Yvette, France, [40]) and a peroxidase-conjugated goat anti-rabbit IgG secondary antibody (1:1000 dilution; GE Healthcare, Milan, Italy) were used to detect *E. focardii* γ-tubulin.

### 2.9. Chemicals, Materials, Reagents and Statistical Analyses

Taq polymerase, RNaseA, DNA modifying and restriction enzymes were purchased from Fermentas (Milan, Italy). Oligonucleotides were synthesized by Sigma/Genosys (Milan, Italy). All routine chemicals were of analytical grade and supplied by Sigma Aldrich (Milan, Italy). All statistical analyses were performed using Microsoft Excel 10.

## 3. Results

### 3.1. EFBTU3 Distinctly Localizes at the Bases of Cilia in E. focardii Cells

A previous work [5] showed that EFBTU3 transcription is stimulated by deciliation, suggesting that EFBTU3 might be involved in cilia regeneration. To verify this hypothesis, we first checked the subcellular localization of EFBTU3 in *E. focardii* cells by immunofluorescence microscopy. We stained the cells with rabbit polyclonal antibodies specifically directed against the isotype-defining C-terminal domain sequence of EFBTU3 (as described under Materials and Methods). For comparison, we also analyzed the subcellular localization of EFBTU1 and EFBTU2, which are transcribed mainly in the later event of cilia recovery [5]. We used antibodies directed against the EFBTU1 and EFBTU2 C-terminal domains, which are identical; thus, both are recognized by the same antibodies. However, the EFBTU2 transcription level is five-fold higher than that of EFBTU1 under physiological conditions (Appendix A); thus, the most representative tubulin upon antibody staining is most likely the EFBTU2. Figure 1 shows confocal microscopic images of *E. focardii* cells stained with a commercial monoclonal antibody directed against the β-tubulin (in green in Figure 1A,E,H,L) and combined with the two polyclonal antibodies directed against either EFBTU2/EFBTU1 (in red, Figure 1B,I) or EFBTU3 (in red, Figure 1F,M). Both monoclonal anti-β-tubulin antibodies and antibodies directed against EFBTU2/EFBTU1 label the intracytoplasmic network and the microtubules of all ciliary structures: (i) adoral membranelles (am) in the cytostomal ciliature, (ii) paraoral membrane (pm) that surrounds the cytostomal area, (iii) four groups of locomotory cirri [fronto-ventral (fv), transverse (tc), caudal (cc) and marginal (mc)] (Figure 1D), and (iv) non-motile cilia of the dorsal surface, which are arranged in longitudinal rows named kineties (ki) (Figure 1K). In contrast, the antibody directed against EFBTU3 recognizes the microtubules at the bases of cilia and cirri but does not accumulate in the elongated cilium (Figure 1F,M).

To confirm the subcellular localization of EFBTU3 in E. focardii cell, we also examined the distribution of EFBTU2/EFBTU1 and EFBTU3 in cellular subfractions enriched in cilia, basal bodies (BB) and cytoplasmic microtubules (cyt) by Western blot, using the monoclonal anti-β-tubulin, and the specific antibodies directed against EFBTU2/EFBTU1 and EFBTU3 (Figure 2). The anti-EFBTU2/EFBTU1 antibodies recognized these isotypes in all three samples as the anti-β-tubulin with a significant enrichment in the cilia, which represent a prominent fraction of the cytoskeletal microtubules (Figure 2, first and second panels). In contrast, the anti-EFBTU3 shows a stronger signal in the fraction containing the basal bodies and a mild accumulation in the cilia sample (Figure 2, third panel), but no apparent bands appear in the sample lane containing the cytoplasmic microtubules. The basal bodies enrichment in the BB fraction was confirmed by using basal bodies-specific anti-γ-tubulin antibodies (Figure 2, fourth panel). To confirm the specificity of the anti-EFBTU3 antibody, we performed the same Western blot analysis by adding a peptide corresponding to the C-terminal domain of EFBTU3 isotype in the immune-recognition reaction. As shown in Appendix A, upon incubation with anti-EFBTU3 antibodies plus the peptide, no bands appeared in the sample containing the basal bodies. Collectively these results indicate that EFBTU3 is mainly incorporated in the microtubules at the bases of the cilia.

### 3.2. EFBTU3-Depleted E. focardii Cells Show Defects in Cell Division and Cilia Regeneration

To get insights on the role of EFBTU3 in *E. focardii* cells, we used RNA interference (RNAi)-mediated gene silencing technology, also known as gene interfering by small double-stranded RNA (siRNA), to knockdown the expression of EFBTU3 in this ciliate. There are no effective strategies to genetically manipulate *E. focardii* yet. However, the use of siRNA to inhibit gene expression has proven to be a powerful tool to analyze the biological function of genes in various eukaryotic organisms [42,47,48,49]. Moreover, this technique has been successfully employed to study genes in several ciliates such as *Stylonychia lemnae* [50], *Paramecium tetraurelia* [41] and also *Euplotes octocarinatus* [51]. The RNAi gene silencing strategy described in ciliates relies on the use of bacteria expressing double-stranded RNAs specific to the gene of interest as a food source. The expression of the siRNAs was induced by pre-incubating the bacteria with IPTG. Two *E. focardii* cultures were fed in parallel with bacteria carrying the construct to produce EFBTU3-specific siRNAs, which promote the degradation of the corresponding transcripts, or with bacteria not containing the construct as control. To test whether the levels of the EFBTU3 mRNAs decrease upon RNAi-silencing, we performed qRT-PCR using, as templates, cDNA samples collected from non-silenced cells (i.e., fed with non-induced bacteria or not carrying the EFBTU3-specific construct, or fed with IPTG-induced bacteria but not carrying the construct) and silenced cells (i.e., fed with IPTG-induced bacteria carrying the EFBTU3-specific construct), following the procedure described under Materials and Methods. We used synchronized *E. focardii* cells for a better understanding of the tubulin silencing effects to avoid, as much as possible, changes due to the different phases of the cell cycle. As shown in Appendix A, EFBTU3 mRNA levels decreased only in the sample obtained from cells fed with IPTG-induced bacteria containing the EFBTU3-specific construct. The reduction of the EFBTU3-specific mRNAs is mirrored by the absence of EFBTU3 at the bases of cilia in silenced cells fixed and stained with anti-EFBTU3 antibodies (Genescript), where instead an unspecific labeling is observed (Appendix A).

After EFBTU3 silencing, we sought for phenotypic defects in *E. focardii* synchronized cells. First, we evaluated the ability of the cells to multiply, since many mutations affecting microtubules impair cell division because cell division requires new ciliogenesis. We monitored the cultures for the ten days of the experiment and counted the number of dividing cells every three days, which is approximately the generation time of *E. focardii* in laboratory conditions [28]. Although the cell morphology of EFBTU3-silenced cells appeared overall identical to that of the control (non-silenced), we observed that the number of cell divisions was reduced in the cell cultures fed with bacteria expressing the EFBTU3-specific siRNAs compared to the non-silenced cells over the ten days of bacteria feeding (Figure 3A): These results suggest that the EFBTU3 supports cell fission (see discussion).

We further investigated the role of EFBTU3 in cilia formation through deciliation of *E. focardii*-synchronized cells fed with bacteria transcribing the EFBTU3 siRNAs and deciliation of cells fed with bacteria not expressing siRNAs as a control. After the deciliation procedure we followed cilia regeneration for the next 4 and 24 h, continuing to feed cells with bacteria. We then fixed and incubated silenced and non-silenced samples with anti-β-tubulin antibodies (as described under Materials and Methods) for immunofluorescence microscopy analysis. As shown in Figure 3, control cells partially and completely regenerated all cilia after 4 h (Figure 3C) and 24 h (Figure 3D) from deciliation, respectively. In contrast, cilia appear not re-formed after 4 h and 24 h from deciliation in EFBTU3-silenced cells (Figure 3F,G, respectively). Both silenced and control cells appear to have well-organized and intact cytoplasmic microtubule structures, demonstrating that the absence of full-formed cilia in silenced cells is due to the RNAi-induced loss of EFBTU3 and not due to the fixing treatment. For a deeper analysis of this result, we collected pictures of 50 cells (randomly chosen in bright field) for each sample after 4 and 24 h from deciliation, and we reported the number of cells showing complete, partial or no regeneration of cilia for each sample in Appendix A. In addition, Appendix A shows cells with complete, partial and no cilia regeneration. We also analyzed the effect of the EFBTU3 silencing on cilia regeneration by Western blot. As shown in Appendix A, the level of EFBTU3 decreased in silenced cells after 24 h of deciliation, whereas the total tubulin pool, EFBTU2 included, is largely not affected. These results confirm that EFBTU3 plays a role in the polymerization of new microtubules during cilia regeneration.

To acquire additional information on the role of EFBTU1 and EFBTU2, we used the same technique to silence the EFBTU2/EFBTU1 isotype. As the target sequence for the RNAi-based silencing of EFBTU2/EFBTU1, we used the sequence corresponding to the EFBTU2 C-terminal domain because, as mentioned earlier, this isotype is five-fold more transcribed than EFBTU1, with an identical C-terminal amino acid sequence and a very similar nucleotide sequence. Therefore, we do not exclude that EFBTU1 is also silenced. In this experiment, the EFBTU2/EFBTU1-silenced *E. focardii* cells died after two days of silencing treatment (data not shown), strongly arguing for an essential role of EFBTU2/EFBTU1 in the formation of the microtubular cytoskeleton. These data further support the hypothesis of distinct roles for *E. focardii* β-tubulin isotypes.

## 4. Discussion

Tubulin isotypes can modulate microtubule dynamics and organization [27,40,52,53,54,55]. For example, high expression of class-III β-tubulin (TUBB3) has been associated with tumoral tissues and low response rates in patients treated with anti-cancer drugs such as taxanes or vinorelbine [56,57,58]. Dominant mutations in TUBB3 alter microtubule function and behavior, causing a class of neurological diseases called “the TUBB3 syndromes” [23]. Furthermore, Rezania et al., demonstrated that the class-III β-tubulin isotype influences the dynamic properties of microtubules composed exclusively by class-II β-tubulin [16]. These data suggest that specific residues in the tubulin primary structure contribute to microtubule diversity and define functional and dynamic properties of tubulin isotypes.

In the Antarctic ciliate *E. focardii*, the EFBTU3 is a divergent isotype with only 87.6% identity to EFBTU2, the most conserved *E. focardii* β-tubulin, and 86.3% identity to the *Euplotes* consensus β-tubulin [30]. Gene expression analysis revealed that EFBTU3 transcription is stimulated by deciliation, suggesting the involvement of EFBTU3 in the first step of cilia regeneration [5]. This property has been confirmed here by EFBTU3 gene silencing. After 24 h from deciliation, only 8% of the silenced cells have completely regenerated cilia, and 34% recovered only partially the ability of cilia regeneration, whereas 58% still did not start cilia regeneration. This result suggests that RNAi strongly affects cilia regeneration. The result that some silenced cells were able to completely regenerate cilia, and that in silenced cells the number of regenerated cilia increased with the increasing of the time from deciliation, may suggest that this process is not totally depending on EFBTU3. It is possible that axonemes are assembled in a longer time with the involvement of the other beta-tubulin isotypes.

The reduced number of cell divisions observed in EFBTU3-silenced cells may be a consequence of the defect observed in cilia formation, since profound cortical changes occur during cell division, and a new ciliogenesis is required [59].

Microtubules are essential components of several cytoskeletal structures, such as cilia, basal bodies and mitotic spindle. Here, we show that EFBTU3 is involved in the formation of only a subset of microtubules: those at the basis of the cilium. This function is in agreement with the fact that EFBTU3 displays a different folding pathway [37] and shows high flexibility in the GTP-EFBTU3 M-loop [5]. These properties suggest that this isotype possesses a peculiar structural conformation that might facilitate the lateral interactions of tubulin heterodimers and chiefly drive the first steps of microtubule polymerization in the cold environment. Therefore, this isotype may favour and/or speed up the polymerization of other tubulin dimers at the bases of cilia, as suggested by the defects in cilia regeneration in *E. focardii* EFBTU3-silenced cells.

It is well known that post-translational modifications play a great role in the control of the microtubule function, similarly to the control of the chromatin function regulated by histone methylation [8,13]. However, all post-translational modifications are added to tubulin subunits only after their incorporation into the microtubule lattice, except for the phosphorylation of Ser172 [60]. The evidence provided here on the specific cellular localization of EFBTU3 isotype in *E. focardii* and its role in cilia regeneration suggest that divergencies in the tubulin primary structure are also relevant.

In conclusion, our results on *E. focardii* indicate that ciliates are valuable models to study the role of tubulin isotypes in microtubule diversity. Furthermore, our results show that RNAi is a suitable technology to better determine the cellular role of proteins when no effective strategies for genetic manipulation of eukaryotic microorganisms are available.

## Figures and Tables

**Figure 1 microorganisms-10-02415-f001:**
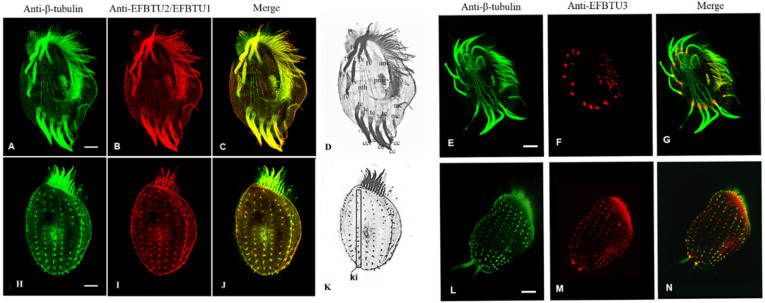
Cellular localization of EFBTU2/EFBTU1 and EFTBTU3 tubulin isotypes in *E. focardii* cells. (**A**–**C**,**E**–**G**) represent confocal immunofluorescence microscopic images of the ventral surface of *E. focardii* cells, whereas (**H**–**J**,**L**–**N**) are images of the dorsal surface of cells co-stained with anti-β-tubulin (green) and either anti-EFBTU2/EFBT1 or anti-EFBTU3 (red) antibodies, respectively. Cytoskeletal structures of the ventral and dorsal surfaces are highlighted in (**D**,**K**), which represents the black and white version of the merged images (**C**,**J**), respectively: adoral membranelles (am), transverse cirri (tc), caudal cirri (cc), marginal cirri (mc) and the microtubule bundles (mb), which elongate from the basal bodies of each transverse cirrus (tc) into the cytoplasm, frontoventral cirri (fv) and kineties (ki). The images in the left panel show that the staining of the isotypes EFBTU2/EFBTU1 largely overlaps with that of the β-tubulin antibodies, which recognize virtually all the microtubular structures of the cell, whereas the right panel shows that the isotype EFBTU3 mainly accumulates in microtubules at the base of the cilium. Scale bars correspond to 10 µm.

**Figure 2 microorganisms-10-02415-f002:**
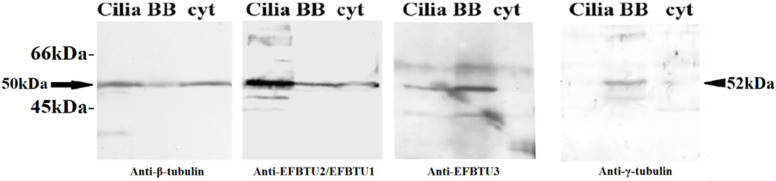
Immunodetection of EFBTU2/EFBTU1 and EFBTU3 in *E. focardii* cellular subfractions by Western blot. Cellular subfractions are enriched in cortical structures including cilia, basal bodies (BB) and cytoplasmic microtubules (cyt). The left panel shows the relative position of the β-tubulin pool (arrow, the estimated molecular weight from the amino acid sequence is nearly 50 kDa for all isotypes) stained by the commercial anti β-tubulin antibodies. In the two central panels, EFBTU2/EFBTU1 and EFBTU3 are stained by anti-EFBTU2/EFBTU1 (left) and anti-EFBTU3 (right) primary antibodies, respectively. *E. focardii* γ-tubulin (arrowhead, the estimated molecular weight from the amino acid sequence is nearly 52 kDa) is shown on the right and is recognized by anti-human-γ-tubulin primary antibody. The molecular weights of protein standards are indicated on the left. The fainter bands are interpreted as products of non-specific recognition by the polyclonal antibodies.

**Figure 3 microorganisms-10-02415-f003:**
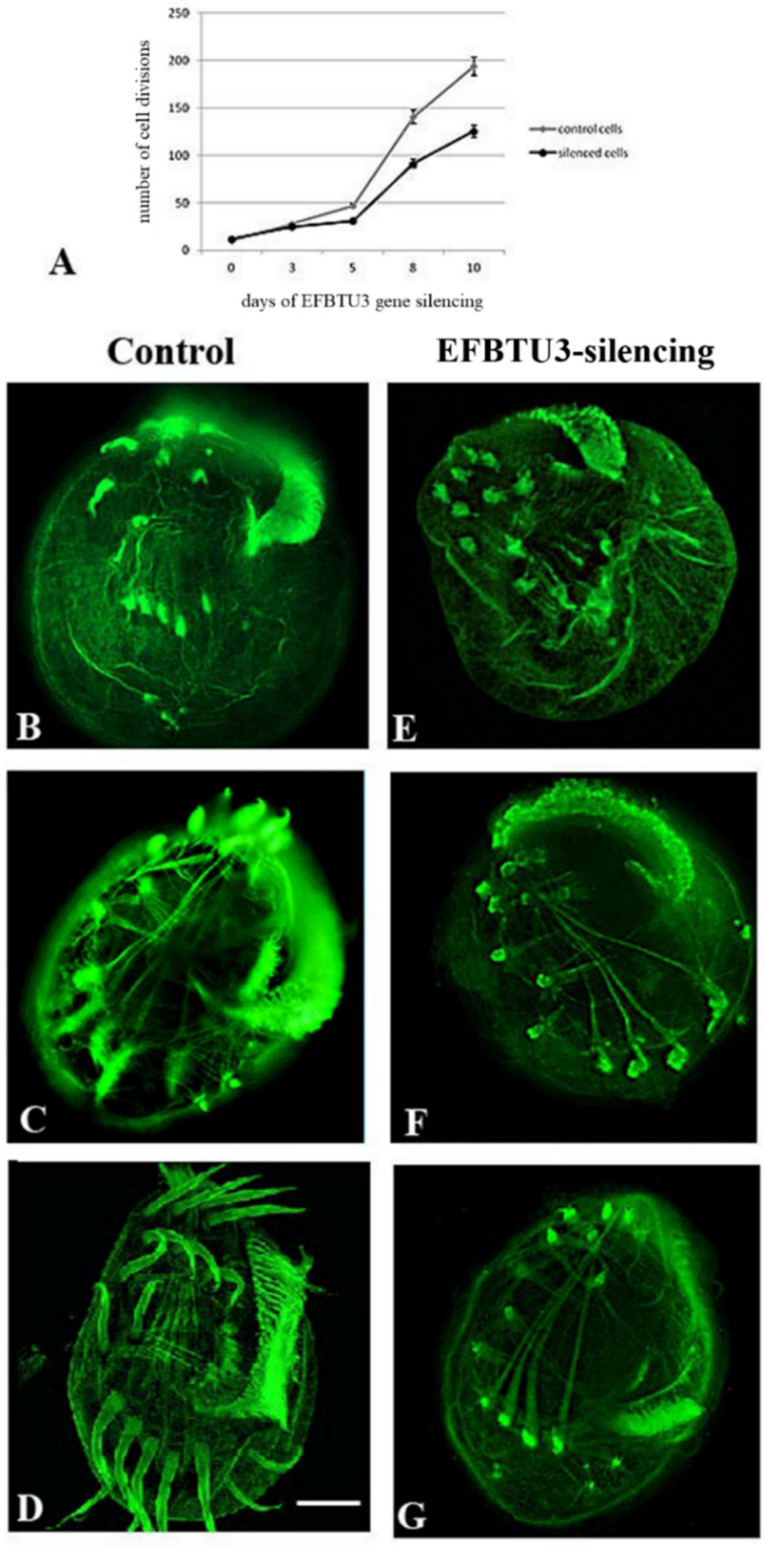
EFBTU3 gene silencing suggests a role for this isotype in cilia regeneration in *E. focardii* cells. (**A**) diagram shows the trends of dividing cells in EFBTU3-silenced cells (fed with bacteria transcribing the EFBTU3 siRNAs, black line) compared to the non-silenced cells (control cells, grey line), during ten days of EFBTU3 gene silencing. (**B**–**G**) show immunofluorescence confocal microscopic images of the ventral region of cells undergone deciliation, collected during a 24-h period. (**B**––**G**) represent control and silenced cells, respectively, collected after 0 (**B**,**E**), 4 (**C**,**F**) and 24 (**D**,**G**) hours from deciliation. *E. focardii* cells were stained with anti-β-tubulin antibodies. In control cells (**B**–**D**), cilia were regenerated within 24 h after deciliation, whereas EFBTU3-silenced cells (**E**–**G**) were not capable to regrow cilia in the same timeframe. Scale bar corresponds to 10 µm.

## Data Availability

Not applicable.

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
