# Peer review of "Ciliate Microtubule Diversities: Insights from the EFBTU3 Tubulin in the Antarctic Ciliate *Euplotes focardii"

_microorganisms, 2022, doi:10.3390/microorganisms10122415_

Round 1

Reviewer 1 Report

MS # microorganisms-2041260 

Title: Ciliate microtubule diversities: Insights from the EfBTU3 Tubulin in the Antarctic Ciliate Euplotes focardii.

by Sandra Pucciarelli et al.

RECOMMENDATION:  A major revision

Comments on the manuscript (MS):

In the present manuscript (MS), the authors use RNA interference (RNAi) to obtain a strain in which the transcript encoding one of the five beta-tubulin isoforms from the ciliate Euplotes focardii (isoform EFBTU3) is destroyed or affected by RNAi, decreasing its transcript levels, and thus obtaining a knock-down strain for this isoform. Likewise, the localization of EFBTU3 is analyzed by immunodetection, and the effects of partial silencing of this isoform on the ciliate corticotype are explored. The work is interesting and well designed, however there are some results that are not sufficiently explained by the authors and some points that should be corrected or clarified.  The most important ones are detailed below:

1- The nomenclature used by the authors to identify the different isotypes of the E. focardii beta-tubulins should be unified, since in the title of the MS it appears as EfBTU3, then in the text both EFBTU3 and EFBT3 are shown (page 4, line 186, page 5, line 196 and so on). The same is true for the isoforms EFBTU1 and EFBTU2. In addition, and as the authors indicate, the EFBTU1 and EFBTU2 isoforms are very similar (98% identity), making it difficult to distinguish between both, and thus the authors indicate it as EFBTU1/EFBTU2. This should be maintained throughout the text, since sometimes only EFBTU2 is indicated (omitting EFBTU1).

2- The molecular masses of each of these 5 isoforms should be indicated, to locate the bands in the western-blots with greater precision, and certain positive bands that appear in the western-blots of which nothing is indicated. Especially considering that the percentages of identity between them is quite high (84% between EFBTU1/2 and EFBTU3). Do they all have the same molecular mass? Since in Figure 2 (anti-EFBTU1/2 and anti-EFBTU3 panels) they seem to have the same molecular mass. Likewise, in Figures 2, S2 and S6 (immunodetection-western-blots, the molecular mass of the positive bands in all panels should be indicated.

3- Page 3 (lines 111 and 112), the authors should indicate either the reference (if published) of the method of culture synchronization or explain with more detail the length of each exposure and the number of times they are exposed to starvation or food. Likewise, the authors should explain the reason, convenience, or necessity of synchronization for this study.

4- As expression levels are quantified from mRNA or transcripts, which are then converted into cDNA, the correct name of the method is qRT-PCR (since a retro-transcription is performed) and not qPCR which uses DNA (useful for analyzing the number of copies of a gene) (page 4, line 141, and elsewhere in the text).

5- In any qRT-PCR analysis, to know if the analysis is reliable and has been well performed, it is necessary to show the parameters of the standard-curves of each of the genes or transcripts analyzed. That is; the efficiency (E%), the Slope value, the value of the correlation coefficient (R2) and the value of y-intercept. These values should be displayed in a table as supplementary material.

6- The sequences of the primers used for the qRT-PCR of EFBTU1 and EFBTU2 are not shown in the manuscript, neither in the text nor in the supplementary material. This is important, considering the results shown in Figure S1, where the transcript levels are very different, but their nucleotide sequences have 96% homology. With such a high level of homology between nucleotide sequences, what kind of primers have been designed and selected to be able to differentiate these two very similar transcripts? This should be clarified by the authors.

It is recommended that all primer sequences used in the different qRT-PCRs be shown in a single table, which could be included in supplementary material.

7- Comparing the histograms of Figures S1 and S3 (both from qRT-PCR analysis), it is observed that in the ordinate axes (vertical axes) the authors use different scales, being in both cases relative abundance of mRNAs. This gives rise to confusion and makes it difficult to compare the real values of the abundance of the different transcripts. These scales should be standardized to indicate the relative abundance of mRNAs for all cases.

Likewise, Figures S1 and S3 should indicate whether or not there are significant differences between the different mRNA abundance levels, and the significant level of these differences.

8- Page 6 (lines 263 and 264). The authors indicate that EFBTU1 and EFBTU2 are mainly transcribed in the later event of cilia recovery. But there is no qRT-PCR (transcription) experiment in the MS to corroborate this statement. If these are previously performed and published experiments, the corresponding reference should be included. In any case, the authors should clarify this sentence.

9- Page 6 (line 270). The source (commercial company, etc.) of the monoclonal antibody against beta-tubulin is not indicated in the materials and methods section. It should be indicated.

10- The legend of Figure 1 does not indicate the scale value that appears on the A, E, H, and L micrographs.

11- Figures 2 and S2. (Figure 2, EFBTU2 and EFBTU3 panels) other positive bands (weaker, below or above the main band) appear. Is this due to possible immunological cross-reactions with other beta-tubulin isoforms? Figure S2 (anti-EFBTU3 panel) also shows a second slightly positive band below the main band.

Something similar occurs in Figure S6, lanes 1 and 2, where another positive band appears above the one indicated by the arrow. The authors should explain the appearance of these secondary positive bands.

Lane 3 in Figures 2 and S2, is indicated as "cyt" in Figure 2 and as "cortex" in Figure S2. To prevent confusion, the nomenclature should be standardized or made uniform.

12- Page 9 (lines 386 and 387). The authors indicate that cells in which EFBTU2 has been silenced die after two days of treatment. But since EFBTU1 and EFBTU2 are very similar in their nucleotide sequence (96%), the authors' reasons for indicating that the silencing is exclusive to EFBTU2 and not also to EFBTU1 should be discussed. And whether the sequences shown in Table 1 for EFBTU2 are specific for that transcript and not for both EFBTU1/2.

13- Figure 3 (panel A). In this graph showing growth kinetics, a growth control with Dunaliella (usual food of this ciliate) is missing. In addition, to better differentiate the growth curves, the parameters of the growth curves should be calculated, i.e., the generation time (Tg) and the growth rate (m). Both can be calculated from the exponential growth phases. 

14- Page 11 (lines 424-426). The authors suggest that the reduced number of cell divisions obtained in the silenced-EFBTU3 culture may be a consequence of the observed defect in the formation of cilia (cirri). However, Table S1 shows that as time goes by (hours) the number of cells that can regenerate cilia (cirri) increases, and the number of cells that do not regenerate cilia (cirri) decreases (see Table S1). The time unit in the growth curves (Figure 3, panel A) is days. And if the values in Table S1 are true, at 3 days (much longer than 24h in Table S1), much of the cell population could have regenerated all the cilia, so it is difficult to attribute the growth delay to the failure to regenerate cilia (cirri). The authors should explain this or look for another hypothesis to explain the growth delay in the silenced strain.

Regardless of this, the authors do not discuss or comment on the results shown in Table S1. The reason that cells that can regenerate cilia (cirri) increase with time (within 24h) should be explained. It is probably because the effect of RNA interference (RNAi) is only effective for a rather limited time. And if so, this should be considered in the interpretation of the results in general. 

15- The discussion section should be extended by resolving the questions raised in the previous points made in the review of this MS. 

In summary, this manuscript is not ready for publication until the questions and comments previously mentioned are resolved and corrected.

Author Response

MS # microorganisms-2041260 

Title: Ciliate microtubule diversities: Insights from the EfBTU3 Tubulin in the Antarctic Ciliate Euplotes focardii.

by Sandra Pucciarelli et al.

We thank very much the reviewers for the insightful and thorough comments. They brought to our attention many details that we did not include in the first version of the manuscript. We produced an improved version (the changes in the revised text are evidenced in yellow) and we answered to all comments here below.

Reviewer n. 1

1- The nomenclature used by the authors to identify the different isotypes of the E. focardii beta-tubulins should be unified, since in the title of the MS it appears as EfBTU3, then in the text both EFBTU3 and EFBT3 are shown (page 4, line 186, page 5, line 196 and so on). The same is true for the isoforms EFBTU1 and EFBTU2. In addition, and as the authors indicate, the EFBTU1 and EFBTU2 isoforms are very similar (98% identity), making it difficult to distinguish between both, and thus the authors indicate it as EFBTU1/EFBTU2. This should be maintained throughout the text, since sometimes only EFBTU2 is indicated (omitting EFBTU1).

Reply: EFBTU nomenclature has been unified through the text and in figures and tables.

The 98% identity between EFBTU1 and EFBTU2 is at amino acid level (and not at nucleotide level) and for this reason, we suppose that the anti-EFBTU2 antibodies may also recognize EFBTU1. Therefore, in the revised version of the manuscript, we carefully mention both isotypes together when we consider the antibody recognition and also the silencing experiment, although we believe that mostly EFBTU2 is the recognised protein as explained in lines 285-287 and also the most silenced protein as explained in lines 408-416. For other analyses, such as the qRT-PCR, we explained why we are confident to distinguish among the two isotypes (legend of Figure S1). 

2- The molecular masses of each of these 5 isoforms should be indicated, to locate the bands in the western-blots with greater precision, and certain positive bands that appear in the western-blots of which nothing is indicated. Especially considering that the percentages of identity between them is quite high (84% between EFBTU1/2 and EFBTU3). Do they all have the same molecular mass? Since in Figure 2 (anti-EFBTU1/2 and anti-EFBTU3 panels) they seem to have the same molecular mass. Likewise, in Figures 2, S2 and S6 (immunodetection-western-blots, the molecular mass of the positive bands in all panels should be indicated.

Reply: The molecular mass estimated from the primary structure of EFBTU1, EFBTU2 and EFBTU3 is 4,8, 4,8, and 4,9 kDa respectively. However, due to posttranslational modifications, it is difficult to have a precise estimation of the final tubulin molecular mass identified in gel electrophoresis. Therefore, we prefer to keep it around 50 kDa and we report this estimation in the corresponding figures and legends.

3- Page 3 (lines 111 and 112), the authors should indicate either the reference (if published) of the method of culture synchronization or explain with more detail the length of each exposure and the number of times they are exposed to starvation or food. Likewise, the authors should explain the reason, convenience, or necessity of synchronization for this study.

Reply: In the revised version of the paper, we reported the more detailed synchronization protocol (lines 134-136) and the reason why we performed it (lines 372-374).

4- As expression levels are quantified from mRNA or transcripts, which are then converted into cDNA, the correct name of the method is qRT-PCR (since a retro-transcription is performed) and not qPCR which uses DNA (useful for analyzing the number of copies of a gene) (page 4, line 141, and elsewhere in the text).

Reply: The correct name of the method, qRT-PCR, has been used through the text

5- In any qRT-PCR analysis, to know if the analysis is reliable and has been well performed, it is necessary to show the parameters of the standard-curves of each of the genes or transcripts analyzed. That is; the efficiency (E%), the Slope value, the value of the correlation coefficient (R2) and the value of y-intercept. These values should be displayed in a table as supplementary material.

Reply: Values of primers standard curves are reported in Table S2 as it is mentioned in the main text in line 183.

6- The sequences of the primers used for the qRT-PCR of EFBTU1 and EFBTU2 are not shown in the manuscript, neither in the text nor in the supplementary material. This is important, considering the results shown in Figure S1, where the transcript levels are very different, but their nucleotide sequences have 96% homology. With such a high level of homology between nucleotide sequences, what kind of primers have been designed and selected to be able to differentiate these two very similar transcripts? This should be clarified by the authors.

It is recommended that all primer sequences used in the different qRT-PCRs be shown in a single table, which could be included in supplementary material.

Reply: In the revised version of the paper, we added the sequence of the oligonucleotides used as primers for EFBTU1 and EFBTU2 qRT-PCR. To distinguish the two isotypes that are nearly identical at protein level, but not at nucleotide level, we used oligonucleotides complementary to portions of the coding region with different codon usage. We added this last information in the revised version of the paper, specifically in the legend of Figure S1. All primers sequences have now been reported in Table S1, as it is mentioned in the main text in lines149, 182.

7- Comparing the histograms of Figures S1 and S3 (both from qRT-PCR analysis), it is observed that in the ordinate axes (vertical axes) the authors use different scales, being in both cases relative abundance of mRNAs. This gives rise to confusion and makes it difficult to compare the real values of the abundance of the different transcripts. These scales should be standardized to indicate the relative abundance of mRNAs for all cases.

Reply: Scales were standardized in both figures.

Likewise, Figures S1 and S3 should indicate whether or not there are significant differences between the different mRNA abundance levels, and the significant level of these differences.

Reply: One asterisk was added to indicate the significant differences with p-value < 0.05 in the Figure S1 and S3 and it was reported in the legends. Lines 15,16 and line 37,38 in the SM.

8- Page 6 (lines 263 and 264). The authors indicate that EFBTU1 and EFBTU2 are mainly transcribed in the later event of cilia recovery. But there is no qRT-PCR (transcription) experiment in the MS to corroborate this statement. If these are previously performed and published experiments, the corresponding reference should be included. In any case, the authors should clarify this sentence.

Reply: This result has been reported in Chiappori et al. 2012. The citation has been included in the correct position lines 277 and 284.

9- Page 6 (line 270). The source (commercial company, etc.) of the monoclonal antibody against beta-tubulin is not indicated in the materials and methods section. It should be indicated.

Reply: The source of the monoclonal antibodies is now reported in material and methods. Lines 221, 222.

10- The legend of Figure 1 does not indicate the scale value that appears on the A, E, H, and L micrographs.

Reply: The sentence “Scale bars correspond to 10 µm” is now reported in figure 1 legend. Line 316.

11- Figures 2 and S2. (Figure 2, EFBTU2 and EFBTU3 panels) other positive bands (weaker, below or above the main band) appear. Is this due to possible immunological cross-reactions with other beta-tubulin isoforms? Figure S2 (anti-EFBTU3 panel) also shows a second slightly positive band below the main band.

Something similar occurs in Figure S6, lanes 1 and 2, where another positive band appears above the one indicated by the arrow. The authors should explain the appearance of these secondary positive bands.

Reply: Additional slightly positive bands are visible mainly in the blots incubated with the polyclonal antibodies that have been synthesized starting from a peptide corresponding to the C-terminal domain of the beta-tubulin isotypes. We are confident that the additional bands derived from non-specific immunoreactions of these antibodies, particularly in Fig 2 and S2. The faint band at the top of gel in Fig S6 may also represent tubulin aggregates, since we used a total cell protein extract. We added these interpretations in the legends. Lines 347, 348. Lines 27, 28 in the SM.

Lane 3 in Figures 2 and S2, is indicated as "cyt" in Figure 2 and as "cortex" in Figure S2. To prevent confusion, the nomenclature should be standardized or made uniform.

Reply: The nomenclature has been standardized. We have used “cyt” in both Figure 2 and Figure S2.

12- Page 9 (lines 386 and 387). The authors indicate that cells in which EFBTU2 has been silenced die after two days of treatment. But since EFBTU1 and EFBTU2 are very similar in their nucleotide sequence (96%), the authors' reasons for indicating that the silencing is exclusive to EFBTU2 and not also to EFBTU1 should be discussed. And whether the sequences shown in Table 1 for EFBTU2 are specific for that transcript and not for both EFBTU1/2.

Reply: By performing RNAi against EFBTU2, we presume that most of the silenced mRNA corresponds to this isotype since we inserted the EFBTU2 specific nucleotide sequence into the L444 vector used to transform E. coli BL21 strain for feeding. In addition, EFBTU1 is lower expressed than EFBTU2. However, we do not completely exclude that some EFBTU1 is also silenced. We better stated this in the revised version of the manuscript. Line 416.

13- Figure 3 (panel A). In this graph showing growth kinetics, a growth control with Dunaliella (usual food of this ciliate) is missing. In addition, to better differentiate the growth curves, the parameters of the growth curves should be calculated, i.e., the generation time (Tg) and the growth rate (m). Both can be calculated from the exponential growth phases. 

Reply: Figure 3 panel A reports the counting of cells that were in the division process during the 10 days of silencing with respect to the not-silenced control cells. Therefore, the figure 3A does not represent growth curves and we believe that an addition of the Dunaliella growth curve could be confusing. The generation time of Euplotes focardii cells has been reported in a previous paper that we have cited in the revision version of the manuscript (Valbonesi and Luporini, 1993). Lines 381-384.

14- Page 11 (lines 424-426). The authors suggest that the reduced number of cell divisions obtained in the silenced-EFBTU3 culture may be a consequence of the observed defect in the formation of cilia (cirri). However, Table S1 shows that as time goes by (hours) the number of cells that can regenerate cilia (cirri) increases, and the number of cells that do not regenerate cilia (cirri) decreases (see Table S1). The time unit in the growth curves (Figure 3, panel A) is days. And if the values in Table S1 are true, at 3 days (much longer than 24h in Table S1), much of the cell population could have regenerated all the cilia, so it is difficult to attribute the growth delay to the failure to regenerate cilia (cirri). The authors should explain this or look for another hypothesis to explain the growth delay in the silenced strain.

Regardless of this, the authors do not discuss or comment on the results shown in Table S1. The reason that cells that can regenerate cilia (cirri) increase with time (within 24h) should be explained. It is probably because the effect of RNA interference (RNAi) is only effective for a rather limited time. And if so, this should be considered in the interpretation of the results in general. 

Reply: We have evidence that RNAi is effective when the feeding with bacteria with L444 vector goes on, even though the silencing is not 100% efficient. We followed the cilia regeneration (this process involves cirri and adoral membranelles that are both very important for cell feeding) for 24 hours because all cilia are completely regenerated in that time in the controls. In silenced cells, after 24h, we still have 58% of cells without any regeneration and 34% with only partial regeneration. We do not expect that in a longer time all cells will completely regenerate the cilia, because most of them will not be able to take food without cilia (adoral membranelles for filtering and cirri for moving toward the food), and will not be able to divide, therefore they will die. However, we cannot exclude that in a longer time, gradually, more cells can regenerate the cilia with respect to what we have analysed in 24 hours. In our interpretation, EFBTU3 is important for the efficiency of cilia regeneration and for the new ciliogenesis required during cell division, but we cannot exclude that during a much longer time the axoneme can be gradually assembled with other beta-tubulin isotypes, without the support of EFBTU3.

15- The discussion section should be extended by resolving the questions raised in the previous points made in the review of this MS. 

Reply: We improved the discussion and we added a conclusion paragraph, following suggestions and comments above reported. Lines 453-460, 463, 473, 480 and 482-486.

Reviewer 2 Report

The study on ‘Ciliate microtubule diversities: Insights from the EfBTU3 Tubulin in the Antarctic Ciliate Euplotes focardii’ by Pucciarelli et al., has focused on Ciliate microtubule diversities as useful models to reveal microtubule complexity and the role of tubulin isotypes. To me, it’s an interesting study and the MS is well-structured. However, the manuscript needs some minor improvements before considering for publication:

1. Page-1, line 34-36: use reference.

2. Page-1, line 38: please add more references.

3. Add a paragraph in the introduction part explaining α- and β-tubulin.

4. Page-2, line 54, 56: please add more references.

5. Page 2, line 64: ‘In ciliated protozoa, all microtubule functions are carried out in a single cell’ – explain and add references.

6. Please add some genetical characteristics of Euplotes focardii in the introduction section.

7. Add the sample collection period in the materials and methods part.

8. Please delete full stop (.) from all sub-sections of materials and methods and results parts.

9. Add the specific conclusion of the study.

Overall, good structured MS and enjoyed reading the contents.

Author Response

Response of the Authors

MS # microorganisms-2041260 

Title: Ciliate microtubule diversities: Insights from the EfBTU3 Tubulin in the Antarctic Ciliate Euplotes focardii.

by Sandra Pucciarelli et al.

We thank very much the reviewers for the insightful and thorough comments. They brought to our attention many details that we did not include in the first version of the manuscript. We produced an improved version (the changes in the revised text are evidenced in yellow) and we answered to all comments here below.

Reviewer n. 2

  1. Page-1, line 34-36: use reference.

Reply: We added a reference, line 50, (Avila, 1992).

  1. Page-1, line 38: please add more references.

Reply: We added a reference, line 54, (Goodson and Jonasson, 2018).

  1. Add a paragraph in the introduction part explaining α- and β-tubulin.

Reply: We added a short paragraph as required, lines 61-64.

  1. Page-2, line 54, 56: please add more references.

Reply: We added a reference, line 57,58, (Detrich and Overton, 1986).

  1. Page 2, line 64: ‘In ciliated protozoa, all microtubule functions are carried out in a single cell’ – explain and add references.

Reply: We wrote a clearer sentence, lines 83,84 and a reference has been added (Wloga et al., 2008) line 84.

  1. Please add some genetical characteristics of Euplotes focardii in the introduction section.

Reply: We added some characteristics, lines 99-103.

  1. Add the sample collection period in the materials and methods part.

Reply: We added this information in the revised version of the manuscript, line 132.

  1. Please delete full stop (.) from all sub-sections of materials and methods and results parts.

Reply: We did as suggested.

  1. Add the specific conclusion of the study.

Reply: A short conclusion was added to the manuscript, lines 482-486.

Round 2

Reviewer 1 Report

Most of the suggested changes and modifications have been made, so the manuscript can be considered ready for publication.